# Knowledge, Perception, and Attitudes during the COVID-19 Pandemic in the Peruvian Population

**DOI:** 10.3390/bs13100807

**Published:** 2023-09-28

**Authors:** Jonathan Adrián Zegarra-Valdivia, Brenda Nadia Chino Vilca, Rita Judith Ames Guerrero, Carmen Paredes-Manrique

**Affiliations:** 1Faculty of Health Sciences, Universidad Señor de Sipán, Chiclayo 14001, Peru; 2Psychology School, Universidad Nacional de San Agustín de Arequipa, Arequipa 04001, Peru; bchino@unsa.edu.pe (B.N.C.V.); cparedesma@unsa.edu.pe (C.P.-M.); 3Psychology School, Catholic University of Santa María, Arequipa 04001, Peru; rames@ucsm.edu.pe

**Keywords:** attitudes, COVID-19, health communication, knowledge, perception, public health, primary prevention

## Abstract

Background: Latin American countries have been profoundly affected by COVID-19. Due to the alarming incidence of identified cases, we intended to explore which psychosocial elements may influence poor adherence to the mandatory control measures among the population. Objective: We aimed to assess Peruvians’ knowledge, attitudes, and vulnerability perception during the coronavirus outbreak. Method: We collected data from 225 self-selected participants using a web-based cross-sectional survey. Results: The overall respondents were between 18 and 29 years old (56.8%), female (59.5%), belonged to educated groups, and graduated professionals (69.3%), most of them. Logistic regression showed that Knowledge is highly associated with education (*p* = 0.031), occupation (*p* = 0.002), and age (*p* = 0.016). Our study identified that, although people reported adequate Knowledge by identifying expected symptoms and virus transmission ways in COVID-19 disease. There is a significant perceived susceptibility to contracting the mentioned virus, displaying stigmatized behavior (59.1%) and fear of contracting the virus from others (70.2%). Additionally, it is reported to lack people’s confidence in national health authorities regarding sanitary responses (62.7%), preparedness for the disease (76.9%), and the lack of adequate measures to deal with it (51.1%). Conclusion: We found that age, education, and occupation modulate Knowledge. At the same time, only age affected Perception and Attitude. Public policies should consider specific guidelines on knowledge translation and risk communication strategies for both containing psychological responses promptly and ensuring compliance with general control measures by the population.

## 1. Introduction

In December 2019, a new viral infection emerged in Wuhan, China [1], named novel coronavirus disease (COVID-19) by the World Health Organization [2]. The unknown nature of the virus has led to alarming death rates in many countries worldwide, putting strain on health systems [2,3]. Studies comparing COVID-19 to previous epidemics like SARS or MERS reveal that the virus has a much broader dispersal capacity [4], indicating a higher potential risk and potentially surpassing infection and death rates previously reported [2,4]. According to the World Health Organization, the number of confirmed cases worldwide is around 1.7 million [5]. COVID-19 has rapidly spread across geographical boundaries, prompting various countries to implement public health protocols to control its spread. Social distancing, hand washing, and city lockdowns have been implemented. This critical situation has elicited various reactions among the population, causing anxiety and fear, particularly among those unaffected by the virus [6].

In Latin America, COVID-19 represents an unprecedented challenge as similar viruses like SARS and MERS have not been experienced in the region before. Many Latin American countries find their public healthcare systems unprepared to handle the epidemic. In Peru, the virus’s rapid spread, even among mildly symptomatic or asymptomatic individuals, highlights the need to understand the population’s behavioral responses to the situation.

Limited studies on knowledge and attitudes during epidemics exist in South America. Earlier studies in the region suggest that the population tends to be hesitant in adopting control measures during outbreaks of diseases like chikungunya, zika, and dengue [7,8]. Non-compliance with government measures during these outbreaks was possibly due to the limited impact on some geographical regions with favorable climatic conditions for those mosquito-borne diseases [8,9,10].

In response to COVID-19, countries have imposed strict control measures to prevent mortality rates from escalating. After confirming its first case on 6 March 2020, Peru implemented strategies like social distancing, continuous hygiene practices, and limiting public movement and access to non-essential places [11]. However, despite the mandatory nature of these protective measures, adherence among the population needs to improve, signaling an alarming lack of commitment among certain groups [6,10,12,13].

Studies analyzing attitudes and knowledge about COVID-19 in Hubei, China, show that attitudes towards government containment measures are closely associated with the level of knowledge about the virus [12]. Individuals with higher information and education levels tend to have more positive attitudes toward preventive practices [6,12]. The perception of risk plays a significant role in the commitment to preventive behaviors during global epidemics [6,10,14,15,16,17].

Perception of risk may be influenced by the type of information individuals have. Lack of information or misinformation can be a barrier, increasing the likelihood of infection [14]. However, people’s judgments are often based on their perception of risk rather than actual risk [16]. During the SARS epidemic, psychological responses generated massive distress, leading to “disproportionate” reactions in the population [18].

Experts in Australia found that poor public communication policies during the H1N1 influenza epidemic contributed to mass panic in the population and non-compliance with containment measures [19]. Individual attitudes toward public policies significantly influence the effectiveness of containment measures [6,19]. Despite government efforts, the passive Attitude towards implemented policies continues to impact the population’s health and that of their close relatives.

The lack of knowledge about COVID-19 may mediate increased virus infection rates. Similar cases, like the Ebola virus outbreak, showed that a poor understanding of the disease and its transmission contributed to higher case rates [20]. Knowledge of infection processes and precautions can influence citizens’ adherence to government guidelines.

Systematic reviews stress the importance of educating affected populations to increase their understanding of the disease cycle and facilitate the adoption of preventive measures [10]. However, studies in developed countries like Singapore indicate that citizens may require less information to comply with government measures, suggesting high trust in their leaders [21]. It is crucial to consider potential biases in these studies, as they mainly assess individuals with higher education levels during the epidemic.

Given the lack of previous studies on outbreaks, knowledge, or risk perception in Peru, our survey aims to assess the population’s level of knowledge regarding COVID-19, its symptoms, transmission, and severity. Additionally, we aim to evaluate the perceived risk and seriousness among the Peruvian population and their behaviors in response to the disease.

## 2. Materials and Methods

### 2.1. Participants

This work is a descriptive, cross-sectional study through a web-based survey [22] conducted between 15 March and 3 April 2020. An initial sample of 225 Peruvian individuals was explicitly recruited in the initial period of the lockdown. The mean age was 31.20 ± 10.97, ranging from 17 to 77 years old, and 59.6% were females. The survey questions were adapted and modified from previously published literature regarding viral epidemics [13,15,21,23,24,25,26,27,28], most related to SARS or MERS disease. The test respondents commented that the questions were easily understood, and the average completion time was 10 min. Informed consent was obtained before starting the survey. Respondents were assured that their responses would be confidential and reminded that their participation in the survey was voluntary. Their Knowledge was evaluated against facts published by WHO [29]. This study was conducted using a convenience sampling of the general Peruvian population with internet access. To calculate the sample, we use a statistical tool, G*Power, using an effect size of 0.15, α error probability of 0.01, power (1-β error probability of 0.999 and 4 predictors for regression analysis as the statistical test used [30].

### 2.2. Instrument: Knowledge, Perception, and Response Questionnaire against COVID-19

Subjects responded to 6 sections of the questionnaire: Knowledge about coronavirus (COVID-19) infection, transmission, perception of disease severity, perceived susceptibility, prevention attitudes, and behavioral response to COVID-19 infection. The sequence in which tests were administered was identical for all subjects. This test was previously described in Zegarra-Valdivia et al. [31]. The survey questions were adapted and modified from previously published literature of similar questionaries under the Ebola, Zika, or A(H1N1) epidemic [7,8,10,13]. A group of trained psychologists systematically analyzed various surveys addressing similar themes such as knowledge, attitudes, and perceptions across diverse epidemic scenarios. During the pilot study, 20 respondents who had participated in the online survey were interviewed. The test respondents found the questions to be easily comprehensible, and the average time taken for completion ranged from 10 to 15 min. Participants were guaranteed the confidentiality of their responses and reminded that their involvement in the interviews was voluntary.

In the knowledge assessment section of the questionnaire, a score of 1 was given for each correctly identified symptom of COVID-19. The subsequent knowledge questions (14 items) were posed in which the answers were Yes, No, or Don’t Know. In the transmission section (10 items), a similar scoring was given for each correctly identified transmission mode of COVID-19. A score of 1 was assigned to a correct answer and a value of 0 to an incorrect answer or do not know the response. In the section about the perception of disease severity, participants indicated the seriousness of COVID-19 in their community context and concerning other viral infections, such as influenza. A three-point Likert-type scale (agree, not sure/maybe, and disagree). 

Questions on perception were divided into five parts. The first part explored perceived susceptibility towards COVID-19 (six items), in which participants indicated their level of exposure by either Yes, No, or Don’t Know. A score of 1 was assigned to a correct answer and a value of 0 to an incorrect answer or do not know the response. The second part examined COVID-19-related fear (four items), with answers like the previous one. 

The third and fourth parts, the susceptibility of getting contagious and contagious places, have 10 and 5 items, respectively. Participants select one of 3 possible answers (very likely, probable, and unlikely). The last part has four items (high, middle, low) and measures the probability of different things related to COVID-19. In the section about the prevention attitude (21 items), participants indicated which behavior is more likely to prevent COVID-19. A three-point Likert-type scale (agree, not sure/maybe, and disagree). Finally, behavioral response to COVID-19 infection explores the attitudes and perceptions about quarantine (3 and 6 items, respectively), in which the answers were either Yes, No, or Do Not Know. Each section has a total score. In the case of knowledge sections, a higher score indicates better Knowledge. A higher score in the perception and behavior score indicates increased vulnerability perception. 

Knowledge items have a high score of 24, Attitude has a high score of 31, and perception has a high score of 38; The total score is 93. This study’s maximum total score was 76 (56.88 ± 7.32). Regarding internal consistency, previous research shows a Cronbach’s Alpha of 0.839 (0.82–0.857 IC 95%) on this instrument [31]. Nonetheless, we analyzed the internal consistency of the instrument. The total Alpha of the Cronbach was 0.811 with a range of sub-scales between (0.311–0.794). Sub-scale with reduced consistency was related to the perception of disease severity. 

### 2.3. Ethical Statement

All participants were informed about the aims of this study and gave written informed consent. This study followed ethics guidelines and was approved by the local ethics committee (CEI number 003-2020). All data were collected in an anonymous database.

### 2.4. Data Analysis

The socio-demographic characteristics of the participants included in the study sample were compared with χ^2^ tests. The χ^2^ test compared the percentages of answers. The effect of age, gender, marital status, occupation, and education was assessed with a linear regression analysis using the total punctuation of the six previous sections. Statistical analysis was performed through the SPSS software version 24 (SPSS, Inc., Armonk, NY, USA). Results were significant with * *p* < 0.05 and ** *p* < 0.01.

## 3. Results

### 3.1. Background Characteristics (Table 1)

The study sample included 225 subjects. Most of the study sample was female (*n* = 134), and it is found a statistically significant difference between age groups by gender (*p* < 0.001 **). From the females, six adolescents (17 years old) were considered in the group <18 and included in the analysis regarding the age close to the age of majority and independence, a situation usually seen in Peru where adolescents do not live with their parents under different conditions. More than half of the respondents were between 18 to 29 years old (56.8%). 69.3% of the sample are graduates, single (70.2%), professional (Workers with a university degree), and independent workers (Technical jobs and trades), displaying a similar percent distribution between males and females.

**Table 1 behavsci-13-00807-t001:** Socio-demographic characteristics.

Gender				
Socio-Demographic Characteristics	Male (*n* = 91)	Female (*n* = 134)	*p* Value	All (*n* = 225)
Age Group	% of Males	% of Females		%
<18 years	---	6–4.5%	**0.001 ** ^a^**	6–2.7%
18–24 years	20–22.0%	44–32.8%		64–28.4%
25–29 years	22–24.2%	42–31.3%		64–28.4%
30–34 years	20–22.0%	12–9.0%		32–14.2%
35–39 years	11–12.1%	3–2.2%		14–6.2%
40–60 years	15–16.5%	23–17.2%		38–16.9%
>60 years	3–3.3%	4–3.0%		7–3.1%
Educational level				
High school	4–4.4%	8–6.0%	0.496	12–5.3%
technician	11–12.1%	9–6.7%		20–8.9%
graduate	60–65.9%	96–71.6%		156–69.3%
postgraduate	16–17.6%	21–15.7%		37–16.4%
Marital Status				
Single	62–68.1%	96–71.6%	0.886	158–70.2%
Married	17–18.7%	21–15.7%		38–16.9%
cohabitating	10–11.0%	13–9.7%		23–10.2%
Widower	---	1–0.7%		1–0.4%
Divorced	2–2.2%	3–2.2%		5–2.2%
Occupation				
Student	22–24.2%	44–32.8%	0.323	66–29.3%
Professional ^b^	51–56%	63–47.0%		114–50.7%
Independent ^c^	18–19.8%	27–20.1%		45–20.0%

^a^: Statistically significant difference (*p* < 0.001 **), χ^2^ square test. ^b^: Workers with a university degree, ^c^: Technical jobs and trades.

### 3.2. Knowledge about Symptoms and Transmission Ways of COVID-19 Disease (Table 2)

The sample does not discriminate between the most frequent symptoms of the disease and includes other manifestations. Thus, more than half of the study sample correctly identified the most frequent symptoms like fever (94.7%), fatigue (62.2%), and dry cough (88.9%) along with others as just as sore throat (81.8%), joint and muscle pain (56.9%). A certain consensus is also observed among the subjects in recognizing as a manifestation of the disease the shortness of breath/shortness of breath (92%). However, this has not been confirmed as part of the diagnosis [30]. Diarrhea (64.9%), runny nose (60.9%), and nasal congestion (66.2%) were not recognized as part of the disease despite being more frequent than other symptoms, such as shortness of breath/shortness of breath. Most of the population (86.2%) knew the incubation period.

In the same way, the situations considered means of transmission/spread of COVID-19 include, in order of importance, Touching objects or surfaces that have been in contact with someone who has the virus (92%), going to areas/countries affected by COVID-19 (88.4%), shake hands with someone who has an active case of coronavirus (84.4%) like the most important. Also, subjects identified situations unrelated to contagion: participating in blood transfusions (59.1%) and relating to people in a hospital or emergency room (53.8%). 

**Table 2 behavsci-13-00807-t002:** Knowledge about COVID-19 symptoms and transmission ways.

What are the Most Frequent Symptoms of Coronavirus (COVID-19)?	Yes	No	I Don’t Know
1.- Fever	**94.7 ^a^**	4.9	0.4
2.- Runny nose	27.6	**60.9 ^a^**	11.6
3.- Sore throat	**81.8 ^a^**	11.1	7.1
4.- Joint and muscle pain	**56.9 ^a^**	31.6	11.6
5.- Shaking chills	32.9	**48 ^a^**	19.1
6.- Shortness of breath/shortness of breath	**92 ^a^**	4.9	3.1
7.- Diarrhea	23.1	**64.9 ^a^**	12
8.- Fatigue	**62.2 ^a^**	26.2	11.6
9.- Dry cough	**88.9 ^a^**	7.1	4
10.- Nasal congestion	21.3	**66.2 ^a^**	12.4
11.- Weightloss	9.8	**71.6 ^a^**	18.7
12.- Stomach discomfort	11.1	**72.4 ^a^**	16.4
13.- Difficulty to sleep	16.4	**62.7 ^a^**	20.9
14.- The incubation period is 5–14 days	**86.2 ^a^**	6.2	7.6
**Which of the following situations are means of transmission/spread of coronavirus (COVID-19)?**			
1.- Coughing or sneezing near people infected with the coronavirus (COVID-19)	**73.8 ^a^**	23.6	2.7
2.- Go to areas/countries affected by a coronavirus (COVID-19)	**88.4 ^a^**	9.3	2.2
3.- Touching objects or surfaces that have been in contact with someone who has the virus	**92 ^a^**	4.9	3.1
4.- Shake hands with someone who has an active case of coronavirus (COVID-19)	**84.4 ^a^**	9.8	5.8
5.- Being on the same plane with someone with coronavirus (COVID-19)	**73.3 ^a^**	21.3	5.3
6.- Eating food prepared by someone infected or exposed to the coronavirus (COVID-19)	**64.9 ^a^**	23.1	12
7.- Participate in blood transfusions	16.9	**59.1 ^a^**	24
8.- By relating to people who were in a hospital or emergency room	35.6	**53.8 ^a^**	10.7
9.- Relating to cases identified by doctors	**78.2 ^a^**	15.1	6.7
10.- For relating to cases identified during evaluations at the entry point to my country	**70.2 ^a^**	17.8	12

^a^: Statistically significant difference (*p* < 0.001 **), χ^2^ square test.

### 3.3. The Severity of COVID-19 and Prevention Measures (Table 3)

Regarding the severity of the disease, 91.6% consider COVID-19 as highly contagious, with symptoms like flu and influenza (84.4%). On the other hand, when evaluating the mortality ratio, they do not assess that it is worse than influenza or tuberculosis (76.4%) or causes permanent physical damage to patients (75.1%). However, when comparing the impact of COVID-19 with influenza or the common cold, more than half of the interviewees indicated that the coronavirus would cause a more significant effect (76%). The results also revealed insufficient confidence in the national or local authorities (62.7%), preparedness for the disease (76.9%), and the lack of adequate measures to deal with it (51.1%).

The results evidence an inappropriate understanding of the precautionary measures. At the same time, hand washing has been recognized as the most efficient form of prevention among respondents (98.2%), followed by personal hygiene (97.3%). Conversely, other vital measures were not considered, such as daily temperature control (57.8%) and the use of a mask (59.1%), even though the WHO recommends its use in healthy subjects in combination with frequent hand cleaning [32]. Furthermore, antibiotics are not recognized as the first line of action against the disease (75.1%), a sign of the population’s Knowledge of the treatment.

**Table 3 behavsci-13-00807-t003:** The severity of COVID-19 and prevention measures.

Severity of the Coronavirus (COVID-19). The Coronavirus:					Agree	Not Sure/Maybe	Disagree
1.- It can be cured					**61.8 ^a^**	-----	38.2
2.- It is highly contagious					**91.6 ^a^**	-----	8.4
3.- The coronavirus mortality rate is worse than influenza or tuberculosis					23.6	-----	**76.4 ^a^**
4.- COVID-19 causes permanent physical damage to patients					24.9	-----	**75.1 ^a^**
5.- You have symptoms similar to common flu and influenza					**84.4 ^a^**	-----	15.6
6.- My community/country does not have a coronavirus vaccine					**73.8 ^a^**	-----	26.2
7.- My community/country does not have adequate medicine or treatment for the disease					48.9	-----	51.1
8.- Hospitals in my community/country have not taken adequate infection control measures					38.7	-----	**61.3 ^a^**
9.- Coronavirus impact is worse compared to influenza or common flu					**76 ^a^**	-----	24
10.- The authorities of my country are prepared to face the disease					23.1	-----	**76.9 ^a^**
11.- The response of the health authorities of my country/community is effective					37.3	-----	**62.7 ^a^**
**Knowledge about contagion prevention/precaution measures**							
1.- Washing hands vigorously (soap/water) for 20 s helps prevent/transmit disease					**98.2 ^a^**	-----	1.8
2.- Special care should be taken if a person has coronavirus (COVID-19) symptoms in my community.					**96.9 ^a^**	-----	3.1
3.- Personal hygiene					**97.3 ^a^**	-----	2.7
4.- Healthy lifestyle					**86.7 ^a^**	-----	13.3
5.- Daily temperature monitoring					**57.8 ^a^**	-----	42.2
6.- Avoid traveling abroad.					**90.2 ^a^**	-----	9.8
7.- Use of mask					**59.1 ^a^**	-----	40.9
8.- Clean environment					**90.7 ^a^**	-----	9.3
9.- Stay home if it’s not okay					**88.4 ^a^**	-----	11.6
10.- Seek medical attention if not okay					**91.1 ^a^**	-----	8.9
11.- Avoid crowded places					**98.7 ^a^**	-----	1.3
12.- Separation/isolation of patients with coronavirus (COVID-19)					**97.3 ^a^**	-----	2.7
13.- Sending passengers with coronavirus symptoms (COVID-19) to a hospital or referral center for examination					**77.3 ^a^**	-----	22.7
14.- You used a disinfectant at home or work.					**89.8 ^a^**	-----	10.2
15.- Check symptoms on websites					50.2	-----	49.8
16.- Wore something to clean objects that may have come in contact with someone with coronavirus (COVID-19)					**80.9 ^a^**	-----	19.1
17.- Avoid Asian restaurants or shops					52.4	-----	47.6
18.- Cancel appointments in hospitals or doctor’s offices					52.4	-----	47.6
19.- Avoid public transportation					**87.6 ^a^**	-----	12.4
20.- Antibiotics are the first-line treatment for the management of coronavirus (COVID-19)					24.9	-----	**75.1 ^a^**
21.- Preparation of raw meats and other foods with different knives					23.1	-----	**76.9 ^a^**

^a^: Statistically significant difference (*p* < 0.001 **), χ^2^ square test.

### 3.4. Perceived Susceptibility to COVID-19 (Table 4)

On the other hand, around 59.1% consider that there is a stigma about COVID-19; 72.4% respond to preventive measures to avoid the disease, and 45.8% value that the problems derived from the pandemic will not pass quickly compared to the 35.6% who do not know about it.

One of the greatest fears among the evaluated population is being in contact with people who have returned from abroad (70.2%), followed by eating out (64%), visiting hospitals (63.1%), and having contact with people with flu symptoms (59.6%). Concern for the family is evident (71.6%), considering that one of the groups most susceptible to contagion is the people over 60 years of age (70.2%) in addition to health services personnel (74.7%). Children are considered in the last place of the possible infected subjects (56.4%).

Health institutions (45.8%) and domestic settings (68.4%) are considered places of infectiousness; in addition, the effectiveness of treatments (57.3%) and the effectiveness of available medication or remedies against the disease (75.6%) pose a high-risk vulnerability. 

**Table 4 behavsci-13-00807-t004:** Perceived susceptibility to COVID-19.

Perception and Perceived Susceptibility or Response	Yes	No	I don’t Know
1.- Do you think there is a stigma related to the coronavirus (COVID-19)	**59.1 ^a^**	24	16.9
2.- Thinking that I could become infected with coronavirus (COVID-19) makes me nervous/anxious	**52 ^a^**	42.7	5.3
3.- Nothing I do can stop the risk of catching me	12.9	**72.4 ^a^**	14.7
4.- If I contract the coronavirus (COVID-19), it will have serious consequences for me or my relatives	**74.2 ^a^**	17.3	8.4
5.- I get upset when I think about the coronavirus (COVID-19)	17.8	**76 ^a^**	6.2
6.- Coronavirus (COVID-19) problems will pass quickly	18.7	**45.8 ^a^**	35.6
**Are you afraid of:**			
1.- Fear of being in contact with people with flu symptoms (e.g., cough, runny nose, sneezing, fever)	**59.6 ^a^**	32	8.4
2.- Fear of eating out (for example, street vendor centers, food courts)	**64 ^a^**	32	4
3.- Fear of being in contact with people who have just returned from abroad	**70.2 ^a^**	22.7	7.1
4.- Fear of visiting hospitals	**63.1 ^a^**	32.4	4.4
**Perceived susceptibility to coronavirus infection (COVID-19), Evaluate the possibility of contracting the disease:**	**Very likely**	**Probable**	**Unlikely**
1.- Oneself	12.4	**60.9 ^a^**	26.7
2.- My relatives	18.7	**68.9 ^a^**	12.4
3.- People over 60 years	**70.2 ^a^**	25.8	4
4.- Adults	33.3	**61.8 ^a^**	4.9
5.- Children	23.6	**56.4 ^a^**	20
6.- Medical services personnel	**74.7 ^a^**	22.2	3.1
7.- Food vendors	48.4	47.6	4
8.- Food handlers	44.9	**49.8 ^a^**	5.3
9.- General public	37.3	**59.6 ^a^**	3.1
10.- Taxi drivers	54.7	41.3	4
**Where are people likely to get coronavirus (COVID-19)?**			
1.- Home	16.9	**68.4 ^a^**	14.7
2.- Health institutions	**45.8 ^a^**	40.9	13.3
3.- Public transport	42.2	**43.6 ^a^**	14.2
4.- Markets or shops	19.1	**53.8 ^a^**	27.1
5.- Countries affected by the coronavirus (COVID-19)	4	39.6	**56.4 ^a^**
**What do you think the percentage of:**	**High**	**Middle**	**Low**
1.- Efficacy of treatments for coronavirus (COVID-19)	**57.3 ^a^**	36	6.7
2.- Likelihood of having a major outbreak of coronavirus (COVID-19) from person to person in my community	**71.6 ^a^**	21.8	6.7
3.- Concern that you or your family members will get the virus	**59.1 ^a^**	36.4	4.4
4.- Having effective medications or remedies available	**75.6 ^a^**	15.6	8.9

^a^: Statistically significant difference (*p* < 0.001 **), χ^2^ square test.

### 3.5. Multivariate Analysis and Influence of Sociodemographics in Knowledge, Perception, and Attitudes to COVID-19 (Table 5 and Table 6)

Finally, a multivariate analysis is used to analyze the weight of each proposed variable in the total score. This result shows that Knowledge has a slight but significant correlation with education (*p* < 0.031 *), occupation (*p* < 0.002 *), and age (*p* < 0.016 *) and explains less than 10% of the variance. In the case of perception, occupation (*p* < 0.034 *) has a slightly significant relationship but explains less than 5.2% of the conflict. The remaining variables do not have substantial results (Table 5). Besides, we analyze socio-demographics’ impact (age, sex, marital status, education, and occupation) on the variables studied (Knowledge, Transmission, Severity, Perception, Prevention, and Attitude). We found that age (*p* < 0.001) was a critical positive covariable in Severity, Perception, and Prevention, with a decreased effect on Knowledge. Similarly, for Knowledge, education (*p* = 0.031) and Occupation (*p* < 0.01) show an effect. 

**Table 5 behavsci-13-00807-t005:** Multi-linear regression of summary score by sub-group of questions to verify the model.

	Statistics R2	SEE	F	*p* Value
Knowledge	0.098	1.904	4.734	0.000 **
Transmission	0.031	1.542	1.418	0.219
Severity	0.037	1.325	1.693	0.137
Perception	0.051	4.605	2.344	0.042 *
Prevention	0.039	2.161	1774	0.119
Attitude	0.015	1.721	0.658	0.656

**: Statistically significant difference at *p* < 0.001, *: Statistically significant difference at *p* < 0.05.

**Table 6 behavsci-13-00807-t006:** Results from Multi-linear regression analysis obtained to verify associations with age, education, occupation, and gender.

Variables		Education	Occupation	Age	Gender	Marital Status
**Knowledge**	**Coefficient β**	0.419	−0.627	0.032	−0.225	−0.038
	***p* value**	**0.031 ***	**0.002 ***	**0.016 ***	0.394	0.817
**Transmission**	**Coefficient β**	0.241	−0.169	0.008	−0.124	−0.14
	***p* value**	0.125	0.296	0.47	0.561	0.29
**Severity**	**Coefficient β**	0.23	0.168	**1.00 × 10^−3^**	−0.134	−0.212
	***p* value**	0.089	0.225	0.888	0.466	0.063
**Perception**	**Coefficient β**	−0.14	1.023	**1.10 × 10^−2^**	0.471	0.554
	***p* value**	0.764	0.034	0.741	0.46	0.161
**Prevention**	**Coefficient β**	0.273	0.062	**2.10 × 10^−2^**	0.37	−0.363
	***p* value**	0.215	0.782	0.155	0.216	0.051
**Attitude**	**Coefficient β**	−0.153	0.219	**−3.06 × 10^−5^**	−0.133	0.012
	***p* value**	0.381	0.224	0.998	0.575	0.934

*: Statistically significant difference at *p* < 0.05.

Finally, the scores reached by the sample on knowledge of COVID-19 was 22.40 ± 3.131, where 104 subjects (46%) obtained low knowledge, 72 (32%) medium knowledge, and 49 (21.8%) high knowledge. Regarding perception vulnerability, the medium score was 18.95 ± 4.86, where 56 subjects (24.9%) had low vulnerability perception, 96 (42.7%) had medium vulnerability perception, and 73 (32.4%) had high vulnerability perception. In the case of Attitudes against COVID-19, the medium score reached by the sample was 22.04 ± 3.02, where 83 (36.9%) had low attitudes against COVID-19, 67 (29.8%) had medium levels, and 75 (33.3%) has high levels of Attitude against COVID-19. 

## 4. Discussion

Considering the spread of COVID-19 in Latin American countries and the higher incidence of people infected in Peru, this study aimed to measure the level of Knowledge, perceived vulnerability, and Attitude of the Peruvian population against COVID-19. However, different public health policies and the mandatory nature of these protective measures were implemented in the last months. The adherence of Peruvians to each of them was limited. Previous reports of psychological adherence to protective standards display that level of information and education are related to a positive attitude toward COVID-19 preventive practices [12].

COVID-19 has a higher rate of contagious properties than previous coronaviruses and affects multiple organs. The absence of awareness of hospital infection control and international air travel facilitated rapid global dissemination [33]. In addition, psychological elements such as fear-induced behavior, misinformation, and economic-related concerns would exert significant pressure on the population, limiting compliance with these government measures [34].

At the time this paper was sent for publication, the Peruvian Ministry of Health had reported more than 9.7 thousand cases by COVID-19 infected patients since the first case was reported on March 6, and the total number of deaths is the second highest in Latin America, with 216 [34]. Nonetheless, the behavioral response of Peruvians was not sufficient. Behavioral responses, such as intense fear of infection or coronaphobia [35], are among the most significant indicators in the evaluated sampling. The findings identify that as long as there is Knowledge about dealing with the epidemic, the degree of susceptibility to infection is lower. As described in a study in Pakistan, the failure to follow precautionary measures against pathogens is explained by insufficient Knowledge [14]. 

Given the impact of such magnitude in Latin American contexts, further analysis is suggested to establish better response and epidemic control strategies from the standpoint of the population. Understanding people’s risk perception is critical to ensure efficient health protection practices during virus outbreaks [10].

Regarding Knowledge, perhaps some symptoms are recognized as COVID-19-related (fever, sore throat, shortness of breath), but our participants do not discriminate correctly. Other significant symptoms, such as nasal congestion, runny nose, dry cough, or diarrhea, are usually more frequent in initial states. The incubation period is well recognized in 86% of the population. We found that knowledge was associated with education, occupation, and age, which indicates that people with a higher educational level tend to have greater access to information sources and educational resources, allowing them better to understand scientific and medical materials on the disease. In addition, it highlights that education and occupation may be related to a greater willingness to learn and adapt to new situations, such as the need to understand and act against COVID-19. Age was also associated with knowledge, showing that older people, due to their experience and recollection of past events, may be more aware of infectious disease risks and, therefore, more interested in learning about COVID-19 to protect themselves and their loved ones. 

Routes of transmission of COVID-19 are well recognized (viral droplets in a sneeze, touching infected objects, shaking hands with people infected, etc.). Nevertheless, other medical circumstances were identified, relating increased perception vulnerability to a specific context and medical conditions (for example, 35% believe that COVID-19 spread is related to people in a hospital or emergency room). This affirmation can promote stigma among sanitary personnel. 5–25% of participants are unaware of or recognize transmission routes. Additionally, more than 20% do not realize that being in touch with people identified by doctors is a potential vector of transmission. 

Perception of COVID-19 severity in the community showed that 76.9% believe that the authorities are unprepared to face the disease, and 62.7% think that the authorities’ response is ineffective. This result may be related to less participation in dictated measures by the government, such as social isolation and gender segregation. Different preventive measures are well recognized by participants, such as personal hygiene, washing hands, or a clean environment. Notwithstanding, other practical efforts, such as using a mask (40.9%) or monitoring temperature (42.4%), are not considered. 

Regarding perceived susceptibility, 72.4% believe that “Nothing I do can stop the risk of catching me.” This vulnerable state may be related to poor participation in ineffective measures to avoid contagious, like social distancing or mask faces. On the other hand, 74.2% believe that “If I contracted the coronavirus (COVID-19), it would have serious consequences for me or my relatives”. Despite different epidemiological studies pointing out that mortality is lower than 5% [36,37] even in Peru, current data indicates a mortality rate of over 10%, and recovery is one of the highest in Latin America [34]. Participants evidence an elevated fear of being in contact with others (59–70%), in correspondence of personal susceptibility of getting the infection (over 60%) and a high likelihood of having a significant outbreak of coronavirus (COVID-19) from person to person in my community (71.6%). 

Regarding Multivariate analysis, it shows that educational level and occupation have an impact on Knowledge. Besides, age was the most critical covariable affecting most variables (Knowledge, Transmission, Severity, Perception, and Prevention). In this way, it is shown that age and probably the emotional maturity reached have been better mediators of these variables.

Finally, we concluded that insufficient understanding of COVID-19 seems to mediate unsafe behaviors, affecting effective prevention measures and the failure to reduce the rate of people infected. Moreover, the perception of vulnerability is high towards certain risky behaviors regardless of other possible transmission routes.

## 5. Limitations

This study has some constraints. First, causal inferences may not be established since the methodology is derived from a cross-sectional design.

Second, it is related to the sample. Because the study was only focused on the outbreak of COVID-19, we used a web-based survey method to avoid possible transmission, causing the sampling of our research to be voluntary and conducted by an online system. Given this circumstance, the possibility of selection bias must be considered. 

Additionally, much of the sample have access to the internet connection in their computers or cellphones. Because of this, participants may have higher income or better educational access (more than 85% have graduate and postgraduate studies). Also, in the absence of low-income people, less education is needed to know their responses to the COVID-19 pandemic. The sample size is another limitation, and the current wave of misinformation in social media would affect poorer responses [16]. Third, due to the sudden disaster, we could not assess other socio-psychological conditions of the participants before the outbreak.

## 6. Recommendations

Due to fearful attitudes and the significant impact on population mental health towards the pandemic and new demands for surveillance and control of current COVID-19 outbreaks. Some previous studies identified appropriate suggestions to facilitate compliance with control measures by the population [14,15] and increase knowledge [27], especially enfaces in psychological coping [38,39]. Some of these are described below: First, educational intervention should be tailored to vulnerable communities, including teaching preventive measures and practical identification of risks in non-technical language [40]. The population must be educated to choose wisely regarding reliable news, such as facts and evidence-based data [41]. On the other hand, it is important to consider the knowledge and attitudes toward possible treatments since some studies conclude that the fear of becoming infected with COVID-19 helps the intention to get vaccinated. However, conversely, conspiracy theories about vaccines arise about their effectiveness. Hence, disseminating knowledge that is easy to understand to the population is essential for better reception and greater acceptance of the vaccine [42,43]. Likewise, the population must be educated about the post-COVID-19 syndrome and the possible consequences of the infection. For this, it is important to follow up on patients. However, there are few studies regarding it [44]. Second, consideration should be given to guiding the population on protecting their mental health by limiting the time they are exposed to information related to COVID-19 during the day [45,46], as well as the need to implement preventive actions in the general population to reduce the prevalence of depressive, anxious and fearful symptoms related to COVID-19 [47].

Third, It is crucial to encourage people to return to their usual work and rest schedule as much as possible to mitigate anguish and fear and ensure sleep quality before going to sleep [18,40].

Besides these recommendations, we believe that clear communication between the government and the Ministry of Health with the population is crucial, with relevant preventive-based-evidence programs. Mental health services and health education could be implemented in the communities, including by-phone therapy and emotional care. 

## Data Availability

Not applicable.

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
