# Peer review of "Knowledge, Perception, and Attitudes during the COVID-19 Pandemic in the Peruvian Population"

_behavsci, 2023, doi:10.3390/bs13100807_

Round 1

Reviewer 1 Report (Previous Reviewer 3)

The study of knowledge and perceptions about COVID-19 are regulators of consistent behavior during a pandemic. This (social) knowledge is a regulator of sanitary behavior, and may also be important for the psychological state of an individual. The paper as a whole adequately reflects the results of the conducted research, the results are sufficiently discussed, their limitations related to the sample, the time of the survey and previous studies are formulated.  It should be noted that the article has a certain sociological bias (against psychological).  Nevertheless, its results can be useful for organizing public awareness of the pandemic and developing programs of sanitary behavior in a pandemic.

Author Response

Dear Reviewer,

We would like to express our gratitude for your valuable feedback.

Sincerely,

Reviewer 2 Report (Previous Reviewer 1)

I have read the author's reply and revised version, and agree to accept this article.

I suggest that the author carefully check the references, some of the information is incomplete, missing years, volumes, periods, page numbers and so on.

Author Response

Dear Reviewer,

We would like to express our gratitude for your valuable feedback.

Sincerely,

Reviewer 3 Report (New Reviewer)

Dear Marcia Min,

It is a pleasure for me to review your article titled "Knowledge, Perception, and Attitudes during the COVID-19 Pandemic in the Peruvian Population," which is currently under review for publication in your esteemed journal. I appreciate the opportunity to contribute to the improvement of your work and would like to provide constructive feedback that I hope will be valuable in enriching your study.

Introduction Section: I find it highly relevant that your work addresses the literature on the COVID-19 pandemic in Latin America, which remains limited at present. However, I suggest considering the incorporation of the following recent studies that can contribute to the context of your research:

-          Caycho-Rodríguez, T., Vilca, L.W., Valencia, P.D. et al. Is the meaning of subjective well-being similar in Latin American countries? A cross-cultural measurement invariance study of the WHO-5 well-being index during the COVID-19 pandemic. BMC Psychol 11, 102 (2023). https://doi.org/10.1186/s40359-023-01149-8

-          Caycho-Rodríguez, Tomás, Oré-Kovacs, Nicole, Carbajal-León, Carlos, Llerena-Espezúa, Ximena, Yupanqui-Lorenzo, Daniel E., & Torales, Julio. (2023). Reasons for the use of masks in open areas after the Peruvian government announced that it will no longer be mandatory in the context of COVID-19 in 2022. Medicina clínica y social, 7(2), 61-69. Epub August 00, 2023.https://doi.org/10.52379/mcs.v7i2.289

-          Caycho-Rodríguez T, Tomás JM, Yupanqui-Lorenzo DE, Valencia PD, Carbajal-León C, Vilca LW, Ventura-León J, Paredes-Angeles R, Arias Gallegos WL, Reyes-Bossio M, Delgado-Campusano M, Gallegos M, Rojas-Jara C, Polanco-Carrasco R, Cervigni M, Martino P, Lobos-Rivera ME, Moreta-Herrera R, Palacios Segura DA, Samaniego-Pinho A, Buschiazzo Figares A, Puerta-Cortés DX, Camargo A, Torales J, Monge Blanco JA, González P, Smith-Castro V, Petzold-Rodriguez O, Corrales-Reyes IE, Calderón R, Matute Rivera WY, Ferrufino-Borja D, Ceballos-Vásquez P, Muñoz-Del-Carpio-Toia A, Palacios J, Burgos-Videla C, Florez León AME, Vergara I, Vega D, Shulmeyer MK, Barria-Asenjo NA, Urrutia Rios HT, Lira Lira AE. Relationship Between Fear of COVID-19, Conspiracy Beliefs About Vaccines and Intention to Vaccinate Against COVID-19: A Cross-National Indirect Effect Model in 13 Latin American Countries. Eval Health Prof. 2023 Jul 13:1632787231186621. doi: 10.1177/01632787231186621. Epub ahead of print. PMID: 37439361; PMCID: PMC10345832.

-          Lombardo, M.P., Recalde, O.G., Cervigni, M. et al. Correction to: The Predictive Power and Dominance of Variables of Purpose and Social Support for Depression, Anxiety, and Fear of COVID-19 in Paraguay. Trends in Psychol. (2023). https://doi.org/10.1007/s43076-023-00308-5

-          Vilca, L.W., Chávez, B.V., Fernández, Y.S. et al. Impact of the fear of catching COVID-19 on mental health in undergraduate students: A Predictive Model for anxiety, depression, and insomnia. Curr Psychol 42, 13231–13238 (2023). https://doi.org/10.1007/s12144-021-02542-5

-          Gallegos, M., Martino, P., Razumovskiy, A., Portillo, N., Calandra, M., Caycho-Rodríguez, T. y Cervigni, M. (2022) «Síndrome Post COVID-19 en América Latina y el Caribe: un llamado de atención», Revista Médica de Rosario, 88(3), pp. 114-118. Disponible en: http://www.revistamedicaderosario.org/index.php/rm/article/view/190 (Accedido: 14agosto2023).

-          Caycho-Rodriguez, T., & Gallegos, M. (2022). Vaccination during the COVID-19 pandemic: how to address the complexity of the phenomenon?. CIRUGIA Y CIRUJANOS, 90(6), 860-861.

-          Tomás Caycho-Rodríguez, Pablo D. Valencia, Lindsey W. Vilca, Mauricio Cervigni, Miguel Gallegos, Pablo Martino, Ignacio Barés, Manuel Calandra, César Armando Rey Anacona, Claudio López-Calle, Rodrigo Moreta-Herrera, Edgardo René Chacón-Andrade, Marlon Elías Lobos-Rivera, Perla del Carpio, Yazmín Quintero, Erika Robles, Macerlo Panza Lombardo, Olivia Gamarra Recalde, Andrés Buschiazzo Figares, Michael White & Carmen Burgos Videla (2022) Cross-cultural measurement invariance of the fear of COVID-19 scale in seven Latin American countries, Death Studies, 46:8, 2003-2017, DOI: 10.1080/07481187.2021.1879318

Salazar-Fernández, C.; Baeza-Rivera, M.J.; Manríquez-Robles, D.; Salinas-Oñate, N.; Sallam, M. From Conspiracy to Hesitancy: The Longitudinal Impact of COVID-19 Vaccine Conspiracy Theories on Perceived Vaccine Effectiveness. Vaccines 2023, 11, 1150. https://doi.org/10.3390/vaccines11071150

-          Salazar-Fernández, C., Baeza-Rivera, M. J., Salinas-Oñate, N., & Manríquez-Robles, D. (2023). Should we take care of each other? Enhancing COVID-19 protective behaviors, a study in Chile, Mexico, and Colombia. Journal of Pacific Rim Psychology, 17. https://doi.org/10.1177/18344909231181763

Instrumentation Section: It is crucial to provide a detailed description of the instruments used in your research to enable readers to understand the methodology and validity of your study. Additionally, if your research involves minors, it is important to detail the process of obtaining informed consent and assent, ensuring ethical standards and participant protection.

Use of Scores and Responses: I recommend that, instead of presenting simple "yes" or "no" answers, you consider providing percentages of individuals who chose each option correctly. Additionally, the response format allows for limited variability in the conducted analyses.

Discussion Section: As you develop the discussion, I suggest considering the incorporation of the studies I mentioned earlier. These works could enhance your analysis and provide a more solid context for your conclusions. Furthermore, given the current state of pandemic control, it would be valuable to update your final recommendations and reflect on the contribution your article makes at this point in time.

I appreciate your dedication to research and the effort you've invested in your article. I hope these comments are helpful in improving your work and contributing to the scientific discourse in this field.

I am available for any further inquiries or clarifications you may require.

Best,

Author Response

Dear Reviewer,

We would like to express our gratitude for your valuable feedback. We have enclosed a document containing our corresponding responses.

Sincerely,

This manuscript is a resubmission of an earlier submission. The following is a list of the peer review reports and author responses from that submission.

Round 1

Reviewer 1 Report

Please see the file, Thanks.

Moderate editing of English language required

Author Response

Dear Reviewer, 1

Thanks for your comments. We addressed most of your suggestions. Regarding x2 method, we use the "yes, no, I don't know" responses on each question distribution. We also did not put together the divorced and widowed, because those were specific distributions of responses. We add the CEI number and correct the recommendations.

Furthermore, we add some information for methods (type study, web-based survey method, sampling). Regarding the number of participants, we clarify that these were taken only in an incredibly early period of lockdown. 

We also add Regression analysis was used to covariate age, gender, education, and occupation, and see the effect on Knowledge, Perception, and Attitudes.

We found that age was the most important covariate, with no effect by gender.

Thanks for your comments.

Reviewer 2 Report

comment

This study examined actual conditions of knowledge and attitudes for the COVID-19 in the Peruvian population, using a web-based survey. This manuscript may contribute to this area of research. Further attention to the issues presented below would strengthen the manuscript.

#1
Regarding participants recruitment methods.
In the first part of the method, the method for selecting the target population should be clearly stated so that the reader can judge the representativeness of the target population of the survey. If this study was conducted using a convenience sampling, it should be stated so clearly.
    The specific procedure used for the survey should be stated in the text body. If a web-based survey was conducted, it should be so stated in the Methods section.
    The year in which the survey was conducted should be stated here.
    Table 1 shows that there are only a few people under 18, but is it acceptable to analyze these together with adults? Please state the authors' opinion on the issue.

#2
Regarding Table 1. It would be better to list the number of people together, not just the percentage in the table. What are "Professional" and "Independet" in Occupation respectively? It should be stated regarding the meanings in the text and in the notes to the table.

#3
In the last line of P7. Does the "multi-line analysis" refer to multivariable regression analysis? As these results are considered to be of highly important, it might be better to note it in the table newly created, I believe. It would also be better to interpret the results of the analysis in the Discussion section.

Minor editing of English language required. For instance, in the line 115, perhaps Spanish version of the question mark? I do not have enough knowledge, though.

Author Response

Dear Reviewer, 

Thank you for your comments. We modify our manuscript following your suggestions. About point 1, we add in the methods:

This work is a descriptive, cross-sectional study through a web-based survey (21) conducted between March 15 and April 3 in 2020. An initial sample of 225 Peruvian individuals was explicitly recruited in the initial period of the lockdown. The mean age was 31.20 ± 10.97, ranging from 15 to 77 years old, and 59.6% were females. The survey questions were adapted and modified from previously published literature regarding viral epidemics (14,16,20,22–27), most related to SARS or MERS disease. The test respondents commented that the questions were easily understood, and the average completion time was 10 min. Informed consent was obtained before starting the survey. Respondents were assured that their responses would be confidential and reminded that their participation in the survey was voluntary. Their Knowledge was evaluated against facts published by WHO (28). This study was conducted using a convenience sampling of the general Peruvian population with internet access.

Point 2: we specify the meaning for professional and independent worker, in the next paragraph: 

The study sample included 225 subjects. Most of the study sample was female (n = 134), and it is found a statistically significant difference between age groups by gender (p< 0.001**). From the females, six adolescents (17 years old) were considered in the group <18 and included in the analysis considering the age close to the age of majority and independence. More than half respondents were between 18 to 29 years old (56.8%). 69.3% of the sample are graduates, single (70.2%), professional (Workers with a university degree), and independent workers (Technical jobs and trades), displaying similar percent distribution between males and females.

Point 3: we corrected the multivariable regression model and include two tables to explain better this point with the respective description.

Finally, we verify English writing.

Thanks for your comments.

Reviewer 3 Report

The authors have undertaken an important study of knowledge and ideas about COVID-19, which are of fundamental importance for behavior during a pandemic. This (social) knowledge is a regulator of sanitary behavior, and can also be important for the psychological state of an individual. The article as a whole adequately reflects the results of the conducted research, the results are sufficiently discussed, their limitations related to the sample, the time of the survey and previous studies are formulated.

At the same time, there are several important points that the authors need to pay attention to. 1) in paragraph 2.1, the year of the research should be indicated; 2) it would also be desirable to conduct a pilot comparative analysis, for example, depending on gender, level of education, age (for example, with the help of ANOVA), especially since the data obtained is sufficient for this, thereby establishing more accurate data on behavioral 3) it would also be desirable to highlight in more detail the methodological foundations of comparative analysis, including the role of knowledge and ideas in the regulation of behavior.

I wish the authors success in publishing their work.

Author Response

Dear Reviewer, 3,

Thanks for your comments. We address all your suggestions. First, we add the year of the study, and then we add some multivariate analysis in the study (table 5 and 6). Regression analysis was used to covariables age, gender, education and occupation, and see the effect in the Knowledge, Perception, and Attitudes.

We found that age was the most important covariable, with not effect by gender. 

Thanks for your comments.

Round 2

Reviewer 1 Report

Knowledge, Perception, and Attitudes during COVID-19 Pandemic in the Peruvian Population

1. The template you are using needs to be replaced, not the template of Brain Sciences Journal

2. Line 5-7, please use English ; line 4,  1* should be 1,*

3. Line 12, Knowledge should be knowledge

4. METHOD: Using a web-based cross-sectional survey, we collected data from 225 self-selected participants, evaluating demographic information?  Only evaluate demographic information? No., I suggest you delete this sentence, because this sentence cannot summarize the research content

5. Line 17, 18, Knowledge should be knowledge

6. Line 23-24, We found that age, education, and occupation modulate Knowledge, Perception, and Attitude.,   According to Tables 5-6, I did not find that age, education, and occupation modulate Perception, and Attitude. Why? Please check!

7. Line 17, Please explain what is highly correlated

8. Line 117,  ¿Why ?  line 119-121, Knowledge should be knowledge

9. Most of the content in the Introduction is well written, but I feel that the content is a bit verbose. Please keep it concise and quickly transition to the topic

10. Methods: line 139, How is the reliability and validity of the Instrument?

11. Please supplement the score of the Instrument, such as total score, score range, etc.

12. Please describe the sample size calculation method.

13. Table 1, Diverced should be “divorced”, please modify

14. Table 1, Please use n (%) to present the results, not just %

15. Table 1: Why does the authors analyze the distribution of different sociodemographic characteristics between men and women? Why group by gender?

16. The results section did not present scores for knowledge, attitudes, and perception. How are the levels of these three variables?

17. Line 256-262, please check, according to table 5-6.

18. Line 266,270, table not tabla;   Multi-Linear regression   not  Multi-lineal

19. Please check each data in table 1-6, ensure they are correct

20. I suggest the authors compare the results of this study with similar studies conducted in other countries or populations, Of course, the author will make revisions according to their own time, and if there is no time, it's fine. The purpose of my suggestion is to make the content of this article more in-depth and vivid

21. References are presented poorly, Please check each reference to ensure consistency in format. E.g., 1, 4, 7, 33, 35-36, 38-39,41-42, etc.

22. I suggest the authors carefully revise this article and provide point-by-point responses to ensure that it can be smoothly accepted

minor editing 

Author Response

Dear Reviewer,

We would like to express our sincere gratitude for your valuable suggestions and comments on our article. Your input has helped improve the quality and clarity of the content. We deeply appreciate the time and effort you have dedicated to reviewing our work and providing constructive recommendations. 

We include a point-by-point document with our answers.

Best regards,
